# Effectiveness of a Brief Lifestyle Intervention in the Prenatal Care Setting to Prevent Excessive Gestational Weight Gain and Improve Maternal and Infant Health Outcomes

**DOI:** 10.3390/ijerph19105863

**Published:** 2022-05-11

**Authors:** Franziska Krebs, Laura Lorenz, Farah Nawabi, Adrienne Alayli, Stephanie Stock

**Affiliations:** Institute of Health Economics and Clinical Epidemiology (IGKE), Faculty of Medicine and University Hospital Cologne, University of Cologne, 50935 Cologne, Germany; laura.lorenz@uk-koeln.de (L.L.); farah.nawabi@uk-koeln.de (F.N.); adrienne.alayli@uk-koeln.de (A.A.); stephanie.stock@uk-koeln.de (S.S.)

**Keywords:** maternal health, overweight, obesity, intervention, pregnancy, gestational weight gain

## Abstract

Research on perinatal programming shows that excessive gestational weight gain (GWG) increases the risk of overweight and obesity later in a child’s life and contributes to maternal weight retention and elevated risks of obstetrical complications. This study examined the effectiveness of a brief lifestyle intervention in the prenatal care setting, compared to routine prenatal care, in preventing excessive GWG as well as adverse maternal and infant health outcomes. The GeMuKi study was designed as a cluster RCT using a hybrid effectiveness implementation design and was conducted in the prenatal care setting in Germany. A total of 1466 pregnant women were recruited. Pregnant women in intervention regions received up to six brief counseling sessions on lifestyle topics (e.g., physical activity, nutrition, drug use). Data on GWG and maternal and infant outcomes were entered into a digital data platform by the respective healthcare providers. The intervention resulted in a significant reduction in the proportion of women with excessive GWG (OR = 0.76, 95% CI (0.60 to 0.96), *p* = 0.024). Gestational weight gain in the intervention group was reduced by 1 kg (95% CI (−1.56 to −0.38), *p* < 0.001). No evidence of intervention effects on pregnancy, birth, or neonatal outcomes was found.

## 1. Introduction

Overweight and obesity are major public health concerns. The world health organization (WHO) has identified obesity as a global epidemic and called for urgent public health measures in response to it [1]. Despite this, over 50% of adults and more than 16% of children in Organization for Economic Co-operation and Development (OECD) countries are still overweight or obese [2]. Besides causing multiple health problems in the overall population, maintaining a healthy bodyweight is particularly important in women of childbearing age. In addition to prepregnancy body weight, gestational weight gain (GWG) also plays an important role in terms of maternal and infant health outcomes [3]. Gestational weight gain and infant health are linked through a process known as perinatal programming [4,5]. Excessive GWG is associated with a number of adverse outcomes for both mother and child, such as gestational diabetes mellitus, hypertensive disorders, caesarean sections, being large for gestational age (LGA), macrosomia, childhood obesity, and long-term weight retention in women [6,7,8,9,10,11,12,13,14].

In 1990 and 2009, the National Academy of Medicine (NAM, formerly known as IOM) published recommendations for adequate GWG [15]. However, based on the evidence available, the percentage of pregnant women who gain more than the recommended weight still varies between 47–68.5% across studies and countries [7,8,11,16,17,18,19]. In Germany, 68.5% of pregnant women experience excessive GWG [19]. These numbers illustrate the need for effective preventive measures to reduce the proportion of women experiencing excessive GWG.

Important and potentially modifiable determinants of GWG include maternal health behaviors, such as diet and physical activity [15,20]. Prevention programs to reduce excessive GWG are therefore aimed at modifying these behaviors [21]. Intervention strategies applied in previous prevention programs include dietary counseling, keeping a food diary, weight monitoring, group education on lifestyle topics, and strategies relating to physical activity, such as structured light-intensity exercises and daily walking targets. It is also common to apply a combination of these strategies. Behavior change techniques such as goal setting, reminder messages, and conversational methods such as ‘motivational interviewing’ (MI) are also incorporated into intervention strategies [22]. 

Meta-analyses on the effect of diet and physical-activity-based interventions in reducing GWG indicate significant beneficial effects [23,24,25]. In a meta-analysis of 49 RCTs, diet and/or exercise interventions reduced the risk of excessive GWG by 20% [25]. Two other meta-analyses reported significant mean reductions in total GWG of 1.42 kg [24] and 0.7 kg [23].

However, it is still unclear as to what extent weight gain reductions can be considered clinically important. Evidence on the effects of lifestyle interventions in relation to maternal and neonatal outcomes is inconsistent. In their recent meta-review, Fair et al. reported “some evidence that [...] interventions may reduce the odds of gestational diabetes,” while no effects on other maternal or neonatal outcomes were found [26]. In two other meta-analyses, positive intervention effects on gestational diabetes, macrosomia and LGA [27], and caesarean section rates [23,27] were reported. At the same time, other studies did not find that lifestyle interventions during pregnancy had any effect on any maternal or neonatal health outcomes [28,29].

The International Weight Management in Pregnancy Collaborative Group (i–WIP) has called for lifestyle counseling to be incorporated into routine prenatal consultations as a public health measure to “tackle the obesity epidemic in pregnancy” [30]. Prenatal care settings provide a unique opportunity for lifestyle interventions, as the utilization of prenatal healthcare services by pregnant women in developed countries is high [31,32,33]. Additionally, the results of a systematic review have demonstrated that interventions delivered by healthcare providers during routine prenatal care achieve superior results when compared to interventions that are conducted in other settings and/or by other health professionals (e.g., dieticians, physiotherapists). However, of the 32 studies reviewed, only a small number (*n* = 7) were delivered by healthcare providers in a prenatal care setting, and heterogeneity regarding study populations, calculation of GWG, intervention strategies and effect sizes across these studies was high [34]. Additionally, the review focused exclusively on pregnant women who were overweight or obese.

As adequate GWG reduces the risk of adverse outcomes, including long term weight retention across all body mass index (BMI) categories [9,23,35], further evidence is required on the effectiveness of lifestyle interventions in routine prenatal care settings that target the general pregnant population. In order to bridge these gaps in the current research, an intervention trial was conducted to assess the real-world effectiveness of incorporating a brief lifestyle intervention into routine prenatal care in terms of the impact on GWG and maternal and infant health outcomes.

## 2. Materials and Methods

### 2.1. Trial Design

The GeMuKi trial (acronym for ‘Gemeinsam gesund: Vorsorge plus für Mutter und Kind’—Strengthening health promotion: enhanced check-up visits for mother and child) was designed as a cluster-randomized, controlled trial using a hybrid effectiveness-implementation design (type II) [36]. As such, data on the implementation process for the intervention was collected alongside effectiveness data. Results on the implementation process for the intervention into regular prenatal care will be published separately. A study protocol entailing detailed information on the rationale, design, and methods of the trial has been published previously [37]. In brief, community-based gynecologists, midwives, and pediatricians in the intervention arm of the trial were recruited to conduct the GeMuKi lifestyle intervention during routine prenatal visits and children’s check-ups. Healthcare providers in the control arm provided care as usual. To reduce the risk of contamination, the intervention was allocated at the regional level as opposed to on an individual level.

The trial was conducted in 10 urban and rural regions within the German state of Baden-Wuerttemberg. Two of these regions (one intervention region and one control region) were added at a later stage in order to enlarge the sample frame. The intervention and control regions were paired via propensity score matching, using the average income per capita, birth numbers of BARMER insured persons, and numbers of community-based gynecologists as the matching criteria. The data of BARMER insured persons were used because BARMER was the first insurer to agree to take part in the project. The two regions that were added at a later stage were selected for their comparability with the original regions in terms of these characteristics. The matched study region pairs were subsequently randomized into intervention and control regions.

### 2.2. Participants

The recruitment of pregnant women was conducted by community-based gynecological practices in the trial regions. Broad inclusion criteria were chosen in order to reflect conditions in real-world routine care. Pregnant women were eligible to participate if they were <12 weeks of gestation, ≥18 years of age, had provided written informed consent, possessed proficient German language skills, were insured with a statutory health insurance provider, and were enrolled by one of the participating gynecological practices.

To reduce the risk of bias due to co-interventions, pregnant women who scored highly on the Edinburgh Postnatal Depression Scale (sum score > 9 and/or score = 3 on item 10) were excluded from this trial and were referred to another intervention, which took place simultaneously in the same regions and which targeted stress and anxiety during pregnancy [38].

### 2.3. Lifestyle Intervention Program

The GeMuKi lifestyle intervention program consisted of up to six brief counseling sessions (about ten minutes each) held alongside routine prenatal visits. In Germany, care for pregnant women is primarily provided in the outpatient setting by community-based gynecologists and/or midwives. Regular prenatal appointments provide an ideal setting for preventive measures, as the utilization of prenatal care is high [31,32,39] and they allow for continuous interventions (up to six counseling sessions in six months).

Prior to the start of the field phase, participating healthcare providers in the intervention regions received training on how to deliver the intervention. The lifestyle counseling was conducted using elements of MI. The counseling content was determined in accordance with evidence-based recommendations issued by the German initiative ‘Healthy Start–Young Family Network’ [40]. Lifestyle topics covered during the counseling included physical activity, diet, breastfeeding, and substance use. As part of every counseling session, healthcare providers and pregnant women agreed upon SMART (Specific, Measurable, Achievable, Reasonable, Time-bound) lifestyle goals which could be met by the next counseling session. Following the counseling session, the participating pregnant women received these goals via reminder messages within an app that was specifically designed for the trial. To aid the gynecologists and midwives during the counseling, information on each participant’s previous counseling progress were provided within a web-based data platform, together with sample questions for MI. Information on counseling topics and progress was entered by all the healthcare providers involved, at every counseling session. Details on the GeMuKi lifestyle intervention and digital tools have been published previously [41,42].

### 2.4. Outcomes

The primary outcome of this study was identifying the proportion of women with excessive GWG according to the NAM guidelines of 2009 [15]. Once they had been recruited, the pregnant women filled out a short, paper-based questionnaire to facilitate the collection of their baseline demographic data and prepregnancy weight and height. Data collection in gynecologists’, midwives’, and pediatricians’ practices was carried out via a web-based data platform. The healthcare providers used this data platform to enter information on weight development and complications during check-up visits.

For the primary outcome, GWG was calculated as the difference between the self-reported prepregnancy weight collected at baseline and the weight measured by the gynecologists or midwives during the last prenatal visit. The pregnant women were categorized into four prepregnancy BMI subgroups using WHO cut-off values [43]. Once this was done, each woman’s weight gain was classified as either adequate or excessive, specific to her prepregnancy BMI and gestational age at the time of her last weight measurement, according to 2009 NAM guidelines. As gestational length varies between women, the duration of time over which weight can be gained is different for every participant. Accounting for gestational age at the time of the last weight measurement reduces the risk of misclassification of GWG, and therefore provides the most accurate metric for excessive GWG prevalence [15,44,45]. For this, NAM recommends the following rates of weekly weight gain for the second and third trimesters: 0.44–0.58 kg/week for underweight women; 0.35–0.50 kg/week for women of normal weight; 0.23–0.33 kg/week for overweight women, and 0.17–0.27 kg/week for obese women. For the first trimester, a weight gain of 0.5–2 kg is recommended for all BMI categories [15]. For twin pregnancies, weight gain rates as described by Fox et al. (2010) were applied accordingly, [46] as the 2009 NAM guidelines do not provide weekly weight gain ranges for women carrying twins. In addition to excessive GWG, differences in GWG (measured in kilograms) between the intervention and control groups were also evaluated.

The secondary outcomes discussed in this article cover pregnancy and obstetric and neonatal complications. The healthcare providers recorded information on complications during every check-up appointment using the digital data platform. The outcomes that were considered were: gestational diabetes, hypertensive disorders, bleeding, caesarean sections, preterm birth, being small for gestational age (SGA), LGA, macrosomia, and abnormal 5 min Apgar scores. SGA and LGA were defined as infant birth weight < 10th and >90th percentiles respectively, adjusted for sex and gestational age. Macrosomia was defined as a birthweight > 4000 g, and an abnormal 5 min Apgar score was classified as a score ≤ 6.

Data quality and plausibility was monitored continually throughout the data collection phase. Where data points seemed implausible, the healthcare providers or the pregnant women in question were contacted in order to obtain the correct information.

### 2.5. Sample Size

Sample size was calculated based on the assumption that the intervention would result in a 10% reduction in the proportion of pregnant women who exceeded the gestational weight gain recommendations. This assumption was based on results of previous intervention trials [25]. Further parameters for the sample size calculation included power = 0.80, α = 0.05 and ICC of 0.05. This resulted in a net sample size of 620 pregnant women per group.

### 2.6. Statistical Analyses

The primary and secondary outcomes were compared between the two trial arms using generalized estimating equations (GEEs). This model type was chosen to account for clustering in the data due to the design of the trial. The primary outcome was analyzed by fitting a logistic model, as excessive GWG was coded as a binary variable. Furthermore, to assess differences in the effect of the treatment by prepregnancy BMI category, an interaction model containing a BMI-by-treatment interaction term was run. For continuous outcome data, linear generalized estimating equation models were fitted. Secondary outcomes were analyzed accordingly. The GEE models were specified using an exchangeable working correlation structure and robust standard errors. The adjusted effect sizes and corresponding 95% CIs were calculated, adjusting for prepregnancy BMI category, age, parity, migration status and educational level. All the analyses were performed on an intention-to-treat (ITT) basis. Multiple imputation by chained equations was used to impute missing data, creating 100 imputed datasets. All the analyses were performed using the public domain statistical software R 4.1.2 (http://cran.r-project.org, accessed on 25 November 2021).

The robustness of the results was examined by performing sensitivity analyses. First, a complete case analysis was conducted including only those participants for whom complete data was available. In addition to this, the primary analysis was rerun using inverse probability of treatment weighting (IPTW) as an additional method to account for imbalances in baseline demographic characteristics among women in the intervention and control groups. Imbalances were assessed by calculating standardized mean differences (SMDs). Differences of >0.1 indicate a potential imbalance [47,48]. IPTW eliminates differences between the treatment and control groups by weighting the observations based on their propensity for being treated. Doubly robust estimates were obtained by incorporating the propensity score weights into the outcome regression models.

## 3. Results

### 3.1. Sample Description

A total of 1466 pregnant women were recruited for the trial. After recruitment, 45 women were lost to follow-up due to miscarriage. Another 12 women declined further participation, and 28 women were no longer contactable. The participant flow is depicted in Figure 1.

The demographic characteristics for the sample at baseline are shown in Table 1. The SMDs of the following variables were close to or passed the threshold of 0.1 indicating potential baseline imbalances: prepregnancy BMI (SMD = 0.20), parity (SMD = 0.09), and migration status (SMD = 0.14). To account for these imbalances, all the models were adjusted for the imbalanced variables and only the adjusted results were reported.

### 3.2. Gestational Weight Gain

The results for the primary outcome are shown in Table 2. An estimated proportion of 52.8% of the women in the intervention group and 59.6% of the women in the control group experienced excessive GWG. The results of the adjusted regression analysis showed a significant treatment effect on the proportion of women who had experienced excessive GWG (OR = 0.76, 95% CI (0.60 to 0.96), *p* = 0.024). The estimated prevalence of excessive GWG was highest in the overweight BMI category and lowest in the underweight BMI category. The subgroup analysis yielded a significant treatment effect in women of normal weight only (OR = 0.71, 95% CI (0.52 to 0.97), *p* = 0.031). There were trends for lower proportions of excessive GWG with the intervention in the overweight and obese BMI subgroups, and a higher proportion in the underweight subgroup, though these results did not reach statistical significance.

The estimated mean GWG was 14.2 kg in the control group and 13.3 kg in the intervention group, resulting in a highly significant reduction of 1 kg (95% CI (−1.56 to −0.38), *p* = 0.001) due to the intervention. This effect depended on the prepregnancy BMI category for the women in question. Significant differences in total gestational weight gain between the intervention and control groups were shown in the subgroups for women of normal weight (β = −0.85, 95% CI (−1.57 to −0.14), *p* = 0.019) and overweight women (β = −1.69, 95% CI (2.65 to −0.74), *p* < 0.001), but not in underweight (β = −0.06, 95% CI (−0.77 to 0.65), *p* = 0.873) or obese women (β = −0.65, 95% CI (−2.59 to 1.30), *p* = 0.514). The biggest effect size occurred in the overweight BMI subgroup, with a highly significant mean reduction of 1.7 kg.

### 3.3. Pregnancy, Birth and Neonatal Outcomes

No significant differences were found between the groups for gestational diabetes, hypertension, preterm birth, or birth mode. A trend for a reduction in the rates of bleeding was found, although this result did not reach statistical significance (OR = 0.5, 95% CI (0.23 to 1.10), *p* = 0.084). Similarly, neonatal outcomes did not significantly differ between groups (see Table 3).

### 3.4. Sensitivity Analyses

The effect estimates for the primary outcome obtained from the complete case analysis were comparable to those calculated from the multiply imputed dataset by means of the ITT analysis. Likewise, the IPTW-weighted models and non-weighted models yielded similar results, confirming the validity of the primary analysis strategy (see Appendix A).

## 4. Discussion

Lifestyle interventions delivered by healthcare providers during pregnancy offer the potential to prevent excessive GWG and, in consequence, may improve health outcomes for both mother and child. The results of this study show that a brief lifestyle intervention embedded in routine prenatal care and delivered by prenatal healthcare providers led to a significant reduction in the proportion of women who gained excessive weight according to NAM guidelines. The odds of excessive GWG were reduced by 24% for the women in the intervention group. The subgroup analyses suggested that the treatment effects were only significant in women of normal prepregnancy BMI. Total GWG in the intervention group showed a significant reduction of 1 kg. The greatest reduction in total GWG was found in women in the overweight prepregnancy BMI subgroup, who had a significant reduction of 1.7 kg when compared to the women in the control group. However, the observed decrease in the proportion of women experiencing excessive GWG and a reduction in total GWG were not reflected in the form of evidence for improved pregnancy, birth, or neonatal outcomes.

The results of this trial only provided evidence for intervention effects on excessive GWG in women of normal weight. Women of normal weight represent the largest BMI group among pregnant women in Germany [19]. In conclusion, the intervention could benefit a large number of pregnant women. However, the study did not reveal significant effects regarding excessive GWG in overweight or obese women, the subgroup of women at the highest risk of excessive GWG [49], although a trend for slightly reduced odds was found in the intervention group: by 16% for overweight women and 13% for obese women. Considering the significant reduction in total GWG of 1.7 kg for the overweight women in this study, it can be hypothesized that the intervention was not intense enough for women in this BMI subgroup to achieve an effect on GWG that was large enough to be translated into increased adherence to NAM guidelines. Similarly, in the meta-analysis published by Thangaratinam et al., a significant reduction in GWG of 1.42 kg through interventions was reported in a sample of all BMI categories, without observing the effects on the proportion of women adhering to NAM guidelines [19]. However, every kilogram by which GWG can be reduced should be considered valuable, as GWG is associated with postpartum weight retention and, in the longer term, affects the BMI status of women during subsequent pregnancies [50,51].

As half of women in the GeMuKi sample were primipara, intervention effects on lifestyle changes leading to lower GWG may also be beneficial with regard to the prospect of subsequent pregnancies. Evidence on the sustainability of intervention effects on maternal lifestyle beyond the period of pregnancy is limited; however, initial results from previous studies suggest modest improvements [52,53,54]. The effects of the GeMuKi intervention on dietary and physical activity behaviors during pregnancy and the postpartum period are yet to be published.

This study did not show intervention effects on any of the pregnancy, birth, or neonatal outcomes, which is in line with previous research [23,28,29]. The i-WIP Collaborative Group conducted a meta-analysis of individual participant data that included 12,526 women. The authors found strong evidence for intervention effects on reduced odds of caesarean sections, but not for other pregnancy, birth, or neonatal complications. The authors reported a mean GWG reduction of 0.7 kg with diet and physical-activity-based interventions [23], which is comparable to the effect size found in the GeMuKi study. Evidence on the long-term effects of excessive GWG suggests that it results in a higher risk of overweight and obesity in a child’s later life [8,55,56]. As such, the observed decrease in the proportion of women experiencing excessive GWG and reduction in total GWG are likely to have a positive impact on infant health in the long run, despite the lack of effects in terms of short-term outcomes. Moreover, as the power calculation in the GeMuKi trial was based on the primary outcome, the trial was most likely underpowered in terms of detecting differences in secondary outcomes. Therefore, more RCTs with an adequate sample size need to be conducted in order to determine the effects of lifestyle interventions on short- and long-term maternal and infant health outcomes beyond GWG.

The GeMuKi intervention utilized established structures of routine prenatal care for intervention delivery. Prenatal healthcare providers (e.g., gynecologists and midwives) are particularly well-suited to carrying out the intervention, as they often have a long and trusting relationship with the women in question. However, previous studies reported a lack of knowledge, confidence, and counseling skills on the part of healthcare providers as being barriers to discussing weight and lifestyle-related topics during routine care [57,58]. In the GeMuKi trial, healthcare providers received training on counseling techniques, weight, and lifestyle topics prior to implementation. In addition to this, it became clear that the healthcare providers participating in the GeMuKi trial were particularly interested in lifestyle topics, and were motivated to discuss these during their everyday prenatal care [59]. For lifestyle interventions to be implemented successfully into routine perinatal care on a large scale, strategies for reaching out to gynecologists and midwives across the country and encouraging them to participate in the lifestyle intervention are required. In addition to this, the importance of weight control during pregnancy and lifestyle topics should be incorporated into the education curricula for perinatal healthcare providers. Furthermore, future research should also focus on strategies for reaching underserved and disadvantaged women, as the effects could prove to be even larger in these populations. The participants in the GeMuKi sample were generally well-educated, and migrant women were underrepresented. More research is therefore required in order to identify successful approaches for these populations.

### Strengths and Limitations

The results of this study are drawn from a large, randomized, controlled trial carried out in a routine prenatal care setting, and thus provide real-world evidence. Broad inclusion criteria were deployed in order to permit recruitment of a diverse sample that reflected the general population of women seeking routine prenatal care. Although routine prenatal care theoretically provides an ideal setting in which to reach pregnant women of all status groups, our sample generally consisted of well-educated, middle-class women. This is, to some extent, reflective of the region in which the trial was conducted, but may also be attributed in part to the requirements of the study, which precluded women with insufficient German language skills from participating, for example. As a result, migrant women were underrepresented in the study sample (22.7% in the GeMuKi sample when compared to 33% in the female German population of the same age group [60]). Moreover, it became clear that the intervention and control groups were imbalanced in terms of baseline characteristics such as prepregnancy BMI, parity, and migration status. In cluster RCTs in health services research, the allocation of the intervention is often conducted before the patients can be recruited, for organizational reasons. As both the recruitment and the delivery of the intervention were conducted by healthcare providers, blinding the providers to treatment allocation was not possible. As such, the imbalances in the baseline characteristics very likely reflect a recruitment bias induced by healthcare providers in selecting the patients they deemed the best fit for the intervention. To minimize bias, regression models were adjusted for confounding variables, and an additional IPTW regression approach was applied to support the validity of the primary analysis. Furthermore, the number of counseling sessions completed varied between participants and only a few participants completed the maximum number of six session. More details on the implementation process of the GeMuKi intervention will be published elsewhere.

Another important strength was the MI-based counseling approach, which provided the trial with an established, theory-based technique for facilitating behavioral change [61]. In addition to this, digital intervention components were used to aid the sustainability of the intervention and to simplify research-related processes (e.g., electronic data collection via an app). Digital components have also been shown to be promising intervention tools for vulnerable and hard-to-reach groups, which supports the transferability of the intervention into these populations [62,63,64,65]. Moreover, in the GeMuKi trial, the pregnant women were recruited in an early stage of pregnancy (before the 12th week of gestation) in order to maximize the length of the intervention period. Another of the study’s strengths is the application of NAM weekly weight gain targets in order to determine excessive GWG, which was corrected for gestational age. This reduced the risk of excessive GWG misclassification arising from differences in gestational length. Moreover, this approach meant that the analysis was not restricted to full-term pregnancies only, as would have been the case with the use of total GWG targets also provided by the NAM. Hence, a more stringent ITT approach was applied. Beyond the primary analysis, one limitation of the trial can be seen in the sample size. The results of the subgroup analysis may suffer from a lack of statistical power, as some of the subgroups (underweight women, obese women) only contained a few participants. Likewise, the study was not sufficiently powered to be able to detect differences in secondary outcomes. Another drawback is that the follow-up period was too short to capture changes in long-term health outcomes for mother and child. One-year follow-up results of the study will be prepared for future publications. Lastly, it should be noted that parts of the study were conducted during the COVID-19 pandemic (March 2020–January 2022). Contact restrictions and lockdown measures may have influenced study outcomes independently of the intervention.

## 5. Conclusions

A brief lifestyle intervention delivered by prenatal healthcare providers embedded in routine prenatal care is effective in reducing the prevalence of excessive GWG and GWG, although no evidence for improved maternal and infant health outcomes was found. Excessive GWG places both mother and child at risk of overweight and obesity. As such, lifestyle interventions as part of routine prenatal care offer the potential to promote healthy weight development for multiple generations. Future studies should cover longer follow-up periods in order to evaluate the long-term effects of lifestyle interventions during pregnancy on maternal and infant health. In addition to this, more research should focus on how interventions should be adapted in order to reach underserved and disadvantaged populations. Furthermore, information is required on the processes for implementing lifestyle interventions in routine prenatal care settings, in order to successfully scale up interventions. The GeMuKi trial included a study on implementation processes; this will provide further insights into how healthcare providers and pregnant women have experienced the implementation of the intervention.

## Figures and Tables

**Figure 1 ijerph-19-05863-f001:**
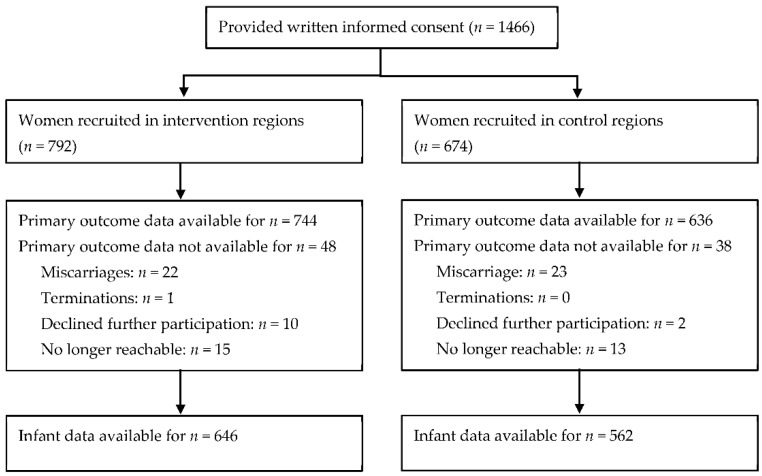
Participant flow.

**Table 1 ijerph-19-05863-t001:** Baseline characteristics of study participants.

	Control Group (*n* = 674)	Intervention Group (*n* = 792)	Total (*n* = 1466)
Age, years	31.3 ± 4.4	31.3 ± 4.3	31.3 ± 4.3
Height, cm	167.0 ± 6.0	166.9 ± 6.1	167.0 ± 6.0
Prepregnancy weight, kg	67.1 ± 14.8	69.8 ± 16.3	68.6 ± 15.6
Prepregnancy BMI, kg/m^2^	24.1 ± 5.2	25.0 ± 5.6	24.6 ± 5.4
Prepregnancy BMI category, *n* (%)			
BMI < 18.5 kg/m^2^	33/674 (4.9%)	20/792 (2.5%)	53/1466 (3.6%)
BMI 18.5–24.9 kg/m^2^	438/674 (65.0%)	477/792 (60.2%)	915/1466 (62.4%)
BMI 25.0–29.9 kg/m^2^	132/674 (19.6%)	172/792 (21.7%)	304/1466 (20.7%)
BMI ≥ 30.0 kg/m^2^	71/674 (10.5%)	123/792 (15.5%)	194/1466 (13.2%)
Parity, *n* (%) nulliparae	345/658 (52.4%)	366/764 (47.9%)	711/1422 (50.0%)
Living with partner	640/667 (96%)	760/780 (97.4%)	1400/1447 (96.8%)
Gestational age at study entry, weeks	9.9 ± 2.0	9.9 ± 1.9	9.9 ± 1.9
Smoker, *n* (%)	18/636 (2.8%)	30/738 (4.1%)	48/1374 (3.5%)
Education, *n* (%)			
Primary	2/645 (0.3%)	0/759 (0.0%)	2/1404 (0.1%)
Lower secondary	19/645 (2.9%)	20/759 (2.6%)	39/1404 (2.8%)
Upper secondary	259/645 (40.2%)	331/759 (43.6%)	590/1404 (42.0%)
University degree	365/645 (56.6%)	408/759 (53.8%)	773/1404 (55.1%)
Immigrant status, *n* (%) immigrants	132/671 (19.7%)	197/776 (25.4%)	329/1447 (22.7%)
First-generation	84/130 (64.6%)	128/194 (66.0%)	212/324 (65.4%)
Second-generation	46/130 (35.4%)	66/194 (34.0%)	112/324 (34.6%)

**Table 2 ijerph-19-05863-t002:** GWG by treatment group.

			Treatment Effect		
	Control Group ^a^	Intervention Group ^a^	Adj. OR (95% CI) ^b^	Adj. Mean Difference (95% CI) ^c^	Adj. *p*-Value
Women exceeding GWG recommendations (total)	59.6%	52.8%	0.76 (0.60 to 0.96)		0.024
BMI < 18.5 kg/m^2^	21.2%	25.8%	1.30 (0.41 to 4.08)		0.605
BMI 18.5–24.9 kg/m^2^	57.5%	48.9%	0.71 (0.52 to 0.97)		0.031
BMI 25.0–29.9 kg/m^2^	81.1%	78.2%	0.84 (0.45 to 1.54)		0.566
BMI ≥ 30.0 kg/m^2^	68.8%	65.6%	0.87 (0.51 to 1.49)		0.658
Total gestational weight gain, kg	14.2	13.3		−0.97 (−1.56 to −0.38)	0.001
BMI < 18.5 kg/m^2^	14.0	14.0		−0.06 (−0.77 to 0.65)	0.873
BMI 18.5–24.9 kg/m^2^	15.5	14.6		−0.85 (−1.57 to −0.14)	0.019
BMI 25.0–29.9 kg/m^2^	15.6	13.9		−1.69 (−2.65 to −0.74)	<0.001
BMI ≥ 30.0 kg/m^2^	11.6	10.9		−0.65 (−2.59 to 1.30)	0.514

^a^ Estimated shares/means. ^b^ Adjusted for prepregnancy BMI, parity, age, migration status, and educational level. ^c^ Adjusted for prepregnancy BMI, parity, age, migration status, educational level, and gestational age at last weight measurement.

**Table 3 ijerph-19-05863-t003:** Pregnancy, birth, and neonatal Outcomes.

			Treatment Effect		
	Control Group ^a^	Intervention Group ^a^	Adj. OR (95% CI) ^b^	Adj. Mean Difference (95% CI) ^b^	Adj. *p*-Value
**Pregnancy and birth outcomes**					
Gestational diabetes mellitus	11.3%	12.4%	1.12 (0.77 to 1.63)		0.537
Dietary treatment	4.0%	4.2%	1.05 (0.55 to 2.02)		0.876
Insulin treatment	2.2% (*n* = 15)	1.9% (*n* = 15)	^c^		^c^
Bleeding	5.1%	2.6%	0.5 (0.23 to 1.10)		0.084
Gestational hypertension	2.4% (*n* = 16)	1.7% (*n* = 13)	^c^		^c^
Preterm birth	7.5%	9.4%	1.28 (0.69 to 2.36)		0.428
Caesarean section	31.6%	35.2%	1.19 (0.86 to 1.64)		0.301
Instrumental delivery	6.9%	7.9%	1.16 (0.68 to 1.96)		0.592
**Neonatal outcomes**					
Birth weight, g	3329.7	3332.1		2.47 (−57 to 61.94)	0.935
Birth length, cm	51.5	51.4		−0.14 (−0.64 to 0.35)	0.572
LGA	5.9%	4.6%	0.76 (0.44 to 1.31)		0.320
SGA	8.5%	8.4%	1 (0.58 to 1.73)		0.993
Macrosomia (birthweight > 4000 g)	10.3%	8.2%	0.76 (0.51 to 1.13)		0.172
Abnormal 5 min Apgar-score (≤6)	2.1% (*n* = 12)	0.5% (*n* = 3)	^c^		^c^

^a^ Estimated shares/means; in cases of small number of cases, no model-based estimations could be obtained and raw shares are displayed in italics. ^b^ Adjusted for prepregnancy BMI, parity, age, migration status, and educational level. ^c^ No statistical modeling due to small number of cases.

## Data Availability

The datasets used and analyzed in this study are available from the corresponding author on reasonable request.

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
