# Peer review of "Effectiveness of a Brief Lifestyle Intervention in the Prenatal Care Setting to Prevent Excessive Gestational Weight Gain and Improve Maternal and Infant Health Outcomes"

_ijerph, 2022, doi:10.3390/ijerph19105863_

Round 1
Reviewer 1 Report
It was my pleasure to review this manuscript that discusses the efficacy of a brief lifestyle intervention in the setting of antenatal care to prevent excessive gestational weight gain and improve maternal and infant health outcomes.
I found the topic very interesting, so I would like to congratulate the authors for having dealt with it.
The introduction seemed correct to me, placing us in the context of the problem and making reference to the relevant literature published so far. The only defect that I would point out is that it contains too many acronyms and makes it a bit difficult for the international scientific community to read since you get easily distracted by not controlling the acronyms well.
The material and method paratado also seemed quite correct, perfectly describing the participants and the interventions that were carried out.
Undoubtedly, the strong point of the manuscript is the regressions that were made to evaluate the results. These results were favorable to prevent weight gain in women who had a normal weight, however, in women who were already overweight prior to pregnancy, the intervention was not significant at all. The existing literature to date seems to support these results.
The discussion also seemed correct to me, justifying the results obtained and clearly expressing the strengths and limitations of the study.
Without a doubt, it seemed to me a very interesting job despite the fact that the results obtained have not been the most hopeful.
Kind regards
Reviewer 2 Report
The article is scientifically sound and interesting to readers.
The schemes are adequate, it is easy to interpret and understand the results.
Reviewer 3 Report
This is an important study, carried out in an empirically sound manner and fully supports the authors findings/conclusions and I think it should be published in its present from after minor spell checking etc.
One small issue I would like clarification on is in the lifestyle intervention section where you state 'using elements of MI' I am assuming that MI is Motivational interviewing? (but the acronym is not introduced in full) I note that the details of the lifestyle intervention are previosuly published but perhaps just clarify here and congratulations on the study.
Reviewer 4 Report
Dear Franziska Krebs and colleagues, Thank you for the submission of your interesting manuscript entitled "Effectiveness of a brief lifestyle intervention in the prenatal care setting to prevent excessive gestational weight gain and improve maternal and infant health outcomes". You have well reported the context for which the intervention was developed, well described study design and applied methodology, clearly presented the results and carefully discussed the results in the light of proclaimed study goals and in the context of similar published studies. In order to emphasise the importance of the evaluated topic and your manuscript I suggest you to mention the significance of physical activity and prevention of prevent excessive gestational weight gain on dysphoric mood and anxiety in late pregnancy (Reference: Petrovic D, Perovic M, Lazovic B, Pantic I. Association between walking, dysphoric mood and anxiety in late pregnancy: A cross-sectional study. Psychiatry Res. 2016 Dec 30;246:360-363. doi: 10.1016/j.psychres.2016.10.009. Epub 2016 Oct 8. PMID: 27770714.). Please provide the full word or phrase for any abbreviation mentioned for the first time in manuscript, for example in Line 31: "of children in OECD countries", the full word or phrase for abbreviation OECD is lacking. Please provide p values for Table 1 entitled (Baseline characteristics of the study sample) in order to demonstrate imbalances observed between the intervention and control groups. I suggest the authors to replace word sample with participants in Title of the table 1).
